# Inactivation of LATS1/2 drives luminal-basal plasticity to initiate basal-like mammary carcinomas

Joseph G. Kern [1], Andrew M. Tilston-Lunel [1], Anthony Federico [2,3], Boting Ning[2,3], Amy Mueller[2,3], Grace B. Peppler[1], Eleni Stampouloglou[1], Nan Cheng[1], Randy L. Johnson[4], Marc E. Lenburg[2,3], Jennifer E. Beane [2,3], Stefano Monti [2,3,5] & Xaralabos Varelas [1] ✉

Basal-like breast cancers, an aggressive breast cancer subtype that has poor treatment options, are thought to arise from luminal mammary epithelial cells that undergo basal plasticity through poorly understood mechanisms. Using genetic mouse models and ex vivo primary organoid cultures, we show that conditional co-deletion of the LATS1 and LATS2 kinases, key effectors of Hippo pathway signaling, in mature mammary luminal epithelial cells promotes the development of Krt14 and Sox9-expressing basal-like carcinomas that metastasize over time. Genetic co-deletion experiments revealed that phenotypes resulting from the loss of LATS1/2 activity are dependent on the transcriptional regulators YAP/TAZ. Gene expression analyses of LATS1/2-deleted mammary epithelial cells notably revealed a transcriptional program that associates with human basal-like breast cancers. Our study demonstrates in vivo roles for the LATS1/2 kinases in mammary epithelial homeostasis and luminal-basal fate control and implicates signaling networks induced upon the loss of LATS1/2 activity in the development of basal-like breast cancer.

Cancers are becoming increasingly understood as heterogeneous populations of cells that evolve over time due to interplay with changing intrinsic and extrinsic environments[1]. Cell state instability allows cancer cells to adapt and favor survival against factors such as changes in the local microenvironment and therapeutic interventions. Mounting evidence suggests that such instability can allow cancers to maintain resident pools of cancer stem cells, which exhibit multi-lineage potential and are thought to facilitate adaptive evolution[2,3]. Cellular plasticity, a phenomenon where cells acquire traits not included in their homeostatic repertoire, is understood as the process by which such phenotypes are adopted by tumor cells. Although cellular plasticity is a mechanism involved in normal processes, such as wound healing, in cancers this process is co-opted as a method for tumor initiation, progression, and therapeutic resistance[4,5].

Basal-like breast cancers are a type of cancer initiated through cell lineage plasticity. These cancers can arise from luminal mammary epithelial cells that acquire basal and stem cell-associated traits upon transformation[5–12]. Sparse treatment options exist for basal-like breast cancers due to a lack of expression of targetable receptors such as the hormone receptors estrogen receptor (ER) and progesterone receptor (PR) and HER2, and their highly heterogeneous transcriptional signatures and mutational landscapes[13]. Improved identification of factors involved in driving luminal-basal plasticity may offer novel avenues for therapeutic intervention of basal-like breast cancers. Blocking luminal-basal plasticity may also improve patient outcomes

[1]Department of Biochemistry, Boston University School of Medicine, Boston, MA 02118, USA. [2]Department of Medicine, Computational Biomedicine Section, Boston University School of Medicine, Boston, MA 02118, USA. [3]Bioinformatics Program, Boston University, Boston, MA 02215, USA. [4]Department of Cancer Biology, University of Texas, MD Anderson Cancer Center, Houston, TX, USA. [5]Department of Biostatistics, Boston University School of Public Health, Boston, MA 02118, USA. ✉e-mail: xvarelas@bu.edu

by inhibiting the acquisition of invasive features, or may allow a reversal of cell fate, rendering basal-like cancers more luminal and thus more easily targeted through established treatment regimens.

The Hippo pathway is a regulator of cell fate in numerous tissues[14]. Previous studies have implicated Hippo pathway components in breast cancer initiation and progression. For example, the Hippo effectors and transcriptional regulators YAP and TAZ (YAP/TAZ) have been shown to permit basal-like traits and aggressiveness in breast cancers[15-19]. Furthermore, the Hippo pathway kinases LATS1 and LATS2 (LATS1/2), which inhibit YAP/TAZ nuclear translocation and transcriptional activity, have been reported to be downregulated in breast cancers[20]. However, the precise roles and cellular phenotypes stimulated by Hippo inactivation in normal mammary cells and breast cancer have yet to be fully understood. We, therefore, sought to better elucidate the impact of Hippo pathway dysregulation in the mammary epithelium in vivo. Here we demonstrate the necessity for LATS1/2 in maintaining luminal cell fate in the mammary epithelium. We show that conditional luminal-specific inactivation of LATS1/2 in the adult mouse mammary epithelium leads to YAP/TAZ-driven luminal-basal plasticity and rapid initiation of basal-like carcinomas that strongly resemble human basal-like breast cancers. Altogether, these findings provide insight into the in vivo roles of the Hippo pathway in mammary epithelial homeostasis and breast cancer initiation and help to clarify mechanisms by which mammary cells undergo plasticity to elicit basal-like cancers.

## Results

### The Hippo pathway kinases LATS1/2 control luminal-basal plasticity in the mammary epithelium

Our analysis of Hippo pathway signaling effectors in the adult mouse mammary epithelium showed distinct levels and localization patterns in luminal and basal cells. When the Hippo pathway is active, LATS1 and LATS2 can be phosphorylated by upstream signaling kinases, such as the MST1/2 kinases, at Thr1079 and Thr1041 (p-LATS1/2), respectively[21]. Analysis of control mouse mammary glands showed high levels and apical localization of p-LATS1/2 in luminal cells, with low detection in basal cells (Fig. 1a). Consistent with previous studies[15,22], we observed that the transcriptional effectors YAP/TAZ showed prominent nuclear localization in basal cells relative to luminal cells (Fig. 1b). These observations suggested high canonical Hippo pathway activity in mature mammary luminal epithelial cells, pointing to a potential role for LATS1/2 in restricting nuclear YAP/TAZ accumulation in this cell lineage.

LATS1/2 function as tumor suppressors in other contexts[23] and luminal mammary cells harbor a potential source for multiple breast cancer subtypes[24], so we sought to test the impact of LATS1/2 inactivation in the luminal mammary epithelium in vivo. For this, we developed a transgenic mouse model that included Lats1 and Lats2-floxed alleles[25,26] and a Tamoxifen-inducible Cre recombinase under the control of the Krt8 promoter (Lats1/2^f/f; K8CreERT2), allowing us to conditionally delete the Lats1 and Lats2 genes specifically in Keratin 8-positive (K8+) luminal epithelial cells[27]. We additionally incorporated an R26-LSL-EYFP lineage trace (Lats1/2^f/f; lsl-EYFP; K8CreERT2), allowing us to visualize cells with Cre activity in these mice. Upon Tamoxifen-induced deletion of LATS1/2 in this model, we observed a dramatic overgrowth of cells within the mammary ducts, leading to a phenotype resembling ductal carcinoma in situ (DCIS) (Fig. 1c). The overgrowth phenotype was observed in every Lats1/2^f/f; K8CreERT2 animal we examined at time points from 2 to 10 days after the last Tamoxifen dose (n > 20 mice). No such DCIS or overgrowth phenotype was observed in control mice. Interestingly, when profiling these cells for markers of luminal (K8) and basal (Keratin 14/K14, Keratin 5/K5) cell fate, we observed EYFP lineage-traced cells expressing K8 and K14 that were negative for K5 expression (K8+, K14+, K5−) expanding within the mammary ducts (Fig. 1d, e and Supplementary Fig. 1a). The acquired

expression of K14 in these luminal-derived carcinomas indicated that loss of LATS1/2 confers markers of luminal-basal fate plasticity to the expanding cell population. In accordance with such plasticity, analysis of luminal EYFP+ lineage traced cells using established mammary epithelial flow cytometry markers[28] demonstrated a reduction in Sca1-expressing cells (Fig. 1f and Supplementary Fig. 1b), a population previously shown to be enriched for hormone receptor expression[28-31]. We further observed a lack of estrogen receptor (ER) and progesterone receptor (PR) expression in carcinomas developing in Lats1/2^f/f; lsl-EYFP; K8CreERT2 mammary glands, phenocopying basal-like breast cancers (Supplementary Fig. 1c, d).

To better validate these findings, we utilized a primary ex vivo organoid culture model with our LATS1/2-floxed mice[32]. We induced luminal-specific LATS1/2 deletion in this system using two methods: (i) infection of Lats1/2^f/f organoids with Ad-K8-nlsCre and (ii) 4-hydroxytamoxifen treatment of organoids grown from mammary cells isolated from Lats1/2^f/f; K8CreERT2 mice. Conditional LATS1/2 deletion in both systems led to dramatic alterations in organoid morphology, including increased organoid size and cellularity (Fig. 1g and Supplementary Fig. 1e–h). When profiling these organoid cultures for expression of luminal and basal-associated genes using RT-qPCR, we found that LATS1/2 deletion led to decreases in the expression of luminal genes and increases in the expression of basal and stem cell-associated genes (Fig. 1h and Supplementary Fig. 1i). The reproducibility we observed across both the Tamoxifen and viral-mediated Cre delivery systems indicated that the observed phenotypes were driven by the loss of LATS1/2 and not from Tamoxifen treatment as a confounding factor. In order to evaluate the mammary-specific progression of carcinomas driven by LATS1/2 loss, we introduced Ad-K8-nlsCre into the mammary ducts of control and LATS1/2-floxed mice harboring an LSL-EYFP or LSL-tdTomato lineage trace[33] using intraductal injections[34] and followed these mice over time. Ad-K8-nlsCre specifically targeted K8+ mammary luminal cells in a control tdTomato mouse (Supplementary Fig. 1j) and when injected into LATS1/2-floxed mice, 12 out of 19 mice developed overt mammary carcinomas over periods of 8–20 months after Cre delivery (Fig. 1i and Supplementary Fig. 1k). 11 of 12 mice that developed primary tumors also showed lineage-traced tumor cell metastases to their lungs (Fig. 1j and Supplementary Fig. 1l), indicating that LATS1/2-deletion-driven mammary carcinomas have invasive and metastatic potential. Histological analysis demonstrated the presence of K8+K14+ tumor cells that also co-expressed the tdTomato lineage trace in these tumors (Fig. 1k). Interestingly, lineage-traced cells within the tumors also expressed high levels of Vimentin and exhibited a mesenchymal morphology (Fig. 1l), suggesting that LATS1/2-deleted cells acquired features of an epithelial–mesenchymal transition.

### Sox9-expressing luminal cells serve as cells of origin for LATS1/2-null carcinomas

The luminal mammary epithelium is comprised of discrete cell lineages, distinguished in part by the expression of ER and the transcription factor Sox9, respectively[35]. Studies in mice have shown that cancers resembling sporadic human basal-like breast cancers can arise from the luminal ER-negative/Sox9-expressing population[7,8]. Human luminal progenitor cells that express high levels of Sox9 are also thought to be the origin of Brca1-driven basal-like breast cancers[10,36]. Our analysis of carcinomas that develop upon luminal LATS1/2 deletion showed widespread expression of Sox9, particularly in expanding K8+K14+ cells (Fig. 2a). Sox9-positive (Sox9+) cells in LATS1/2-deleted carcinomas were also ERα-negative (ER−) (Supplementary Fig. 2a), and when profiling Lats1/2^f/f; lsl-EYFP; K8CreERT2 mice at earlier time points (1–6 days after the last Tamoxifen dose), we observed that Sox9+ cells proliferated and gradually acquired K14 expression in this model (Supplementary Fig. 2b). These results suggested a Sox9+ luminal origin to the basal-like carcinomas that developed. To test this potential

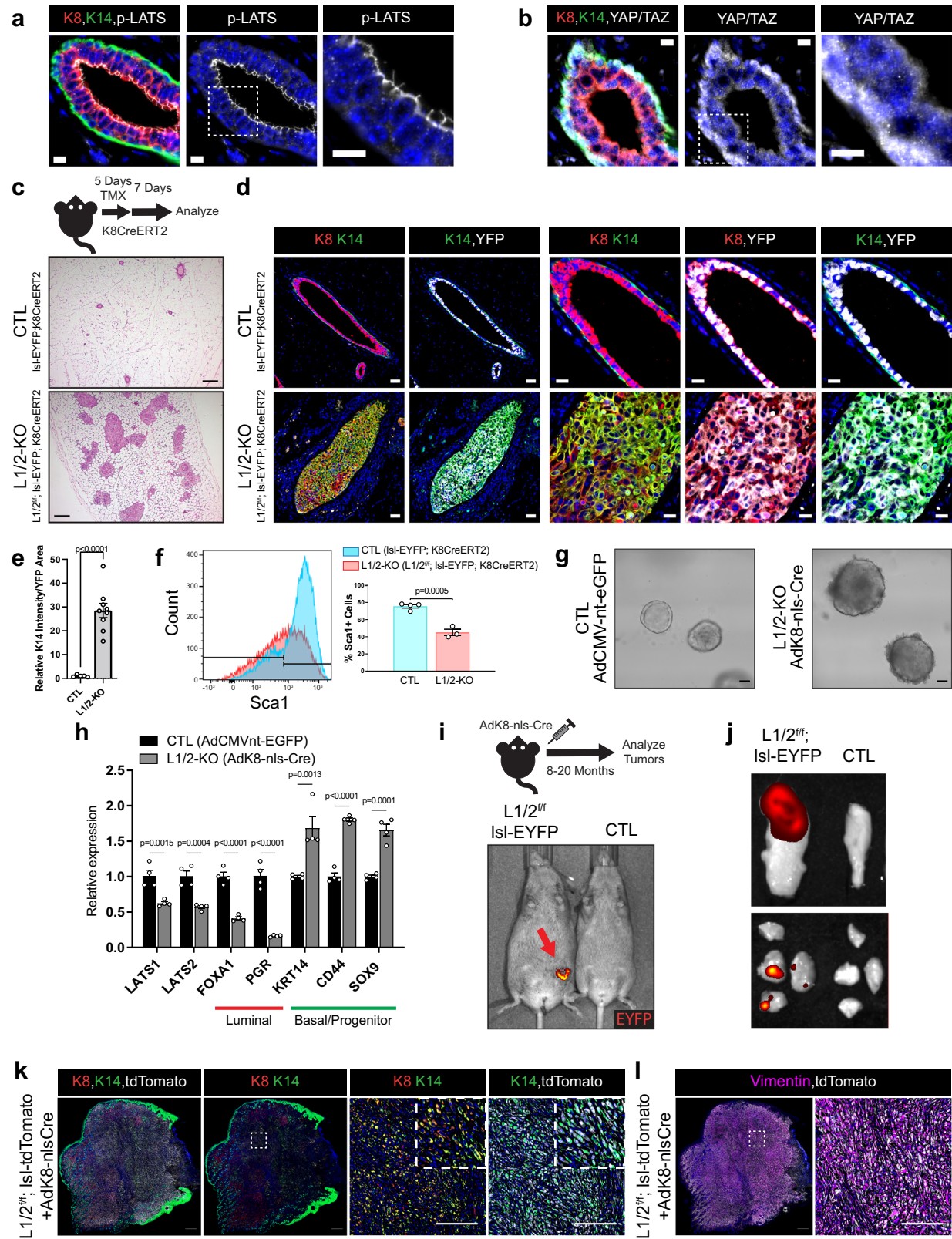

cellular origin, we developed a mouse model to specifically delete LATS1/2 using Tamoxifen-inducible Cre expressed from the endogenous *Sox9* promoter (Sox9CreERT2)[37] combined with an LSL-EYFP-lineage trace (Lats1/2^f/f; lsl-EYFP; Sox9CreERT2). Profiling of LSL-EYFP; Sox9CreERT2 control mice showed high fidelity of the Cre for Sox9⁺ cells (Supplementary Fig. 2c). LATS1/2 deletion in Sox9⁺ cells lead to the rapid expansion of cells within the mammary duct, recapitulating

the phenotype observed with K8CreERT2-driven deletion (Fig. 2b). We observed this phenotype in every animal examined with a Lats1/2^f/f; Sox9CreERT2 genotype at time points from 4 to 13 days after the last Tamoxifen dose (*n* = 14 mice) with no phenotypes being observed in control mice. Like our LATS1/2-floxed; K8CreERT2 model, we observed lineage traced K8⁺K14⁺ cells that lacked the expression of K5 expanding within the mammary ducts of Lats1/2^f/f; lsl-EYFP; Sox9CreERT2 mice

**Fig. 1 | Loss of LATS1/2 in luminal mammary cells induces mammary carcinomas exhibiting luminal-basal plasticity. a, b** Immunofluorescence (IF) staining of **a** phospho-LATS1/2 (Thr1079/1041) or **b** YAP/TAZ together with K8 and K14 in control mouse mammary ducts (Scale bar, 10 µM). **c** Diagram of the mouse model used to conditionally delete LATS1/2 in K8+ luminal mammary epithelial cells along with H&E staining of control and Lats1/2$^{f/f}$;lsl-EYFP;K8CreERT2 mammary glands (Scale bar, 200 µM). **d** IF staining of K8, K14, and EYFP in control and Lats1/2$^{f/f}$;lsl-EYFP;K8CreERT2 mammary glands following tamoxifen treatment (Scale bar 50 µM in the left two columns, and 20 µM in the right three columns). **e** Quantification of K14 intensity/YFP area in mammary ducts of lsl-EYFP;K8CreERT2 (CTL) and Lats1/2$^{f/f}$;lsl-EYFP;K8CreERT2 (L1/2-KO) mice (n = 5 images from 2 CTL mice, n = 9 images from 2 L1/2-KO mice. Unpaired two-tailed t-test, Data are shown as mean ± SEM). **f** Flow cytometric profiling for Sca1 expression in CTL and L1/2-KO mammary epithelial cells (n = 4 CTL, n = 3 L1/2-KO. Unpaired two-tailed t-test, Data are shown as mean ± SEM). **g** Morphology of organoids cultured from control and LATS1/2$^{f/f}$ mice

infected with AdK8-nls-Cre (Scale bar, 50 µM). **h** RNA expression of selected luminal and basal/stem-cell markers in control and LATS1/2$^{f/f}$ organoids infected with AdK8-nls-Cre (n = 4. Unpaired two-tailed t-tests. Data are shown as mean ± SEM). **i** IVIS live imaging of a primary tumor (red arrow) formed in the mammary gland of a LATS1/2$^{f/f}$; lsl-EYFP mouse approximately 14 months after intraductal injection with AdK8-nls-Cre compared to an uninjected control mouse. **j** IVIS imaging of EYFP signal from a representative primary tumor (top) and lung metastases (bottom) formed in a LATS1/2$^{f/f}$; lsl-EYFP mouse approximately 11 months after intraductal injection with AdK8-nls-Cre compared to organs from an uninjected control mouse. **k, l** IF staining of **k** K8, K14, and tdTomato or **l** Vimentin and tdTomato in a primary tumor formed in a LATS1/2$^{f/f}$; lsl-tdTomato mouse approximately 13 months after intraductal injection with AdK8-nls-Cre (n = 2 tumors) (Scale bars, 500 µM on low magnification images on left, and 200 µM on high magnification images of regions outlined by white dotted lines on the right). Source data are provided as a source data file.

(Fig. 2c, d and Supplementary Fig. 2d). These K8+K14+ cells also displayed high levels of Sox9 (Fig. 2e). Organoid experiments performed with Lats1/2$^{f/f}$; Sox9CreERT2 mice showed similar phenotypes to those induced by K8CreERT2, including increases in size (Fig. 2f and Supplementary Fig. 2e, f), decreases in luminal genes, and increases in basal genes (Fig. 2g). Altogether, these results indicated that Sox9+ luminal cells can serve as cells of origin for basal-like carcinomas driven by LATS1/2 deletion, reinforcing the implication of this lineage as an origin for basal-like breast cancers.

Sox9 has been proposed to not only mark cells of origin for basal-like breast cancers but also contribute functionally to basal-like cancer progression and mammary stem and progenitor phenotypes[7,36,38,39]. To explore the functional role of Sox9 in basal-like mammary carcinomas driven by the loss of LATS1/2, we introduced Sox9-floxed alleles[40] in our Lats1/2$^{f/f}$; lsl-EYFP; K8CreERT2 mouse model (Lats1/2$^{f/f}$; Sox9$^{f/f}$; lsl-EYFP; K8CreERT2). Concomitant deletion of LATS1/2 and Sox9 in this model did not reverse the expansion of cells within the mammary ducts (Fig. 2h), as we still observed overgrowth in Lats1/2$^{f/f}$; Sox9$^{f/f}$; K8CreERT2 mice (n = 12) collected 6–7 days after Tamoxifen treatment. Cells expanding within the mammary ducts lacking Sox9 maintained the ability to adopt a K8+K14+ state (Fig. 2i and Supplementary Fig. 2g). However, quantitation of K14 intensity in lineage-traced cells indicated reduced K14 levels in Lats1/2$^{f/f}$; Sox9$^{f/f}$; K8CreERT2 mice relative to Lats1/2$^{f/f}$; K8CreERT2 mice, suggesting that Sox9 contributes to the observed basal-like state observed histologically following LATS1/2 deletion (Fig. 2j).

### Mammary carcinomas driven by LATS1/2 loss are dependent on YAP and TAZ

YAP and TAZ are primary mediators of Hippo pathway signaling that transduce signals controlled by LATS1/2 in many other contexts[23]. To determine whether YAP/TAZ are necessary to produce luminal-basal plasticity and carcinogenesis driven by LATS1/2 deletion, we profiled YAP/TAZ levels in Lats1/2$^{f/f}$; lsl-EYFP; K8CreERT2 mice. K8+K14+ cells in the mammary carcinomas in these mice showed increased nuclear YAP/TAZ levels relative to control K8+ cells (Fig. 3a). Similar elevated nuclear YAP/TAZ was observed in Lats1/2$^{f/f}$; lsl-EYFP; Sox9CreERT2 mammary carcinomas (Supplementary Fig. 3a). To test the necessity of YAP/TAZ in LATS1/2-deleted tumor development, we introduced Yap-floxed and Taz/Wwtr1-floxed alleles[41] in our Lats1/2$^{f/f}$; lsl-EYFP; K8CreERT2 model (Lats1/2$^{f/f}$; YAP$^{f/f}$; TAZ$^{f/f}$; lsl-EYFP; K8CreERT2), allowing us to conditionally co-delete YAP and TAZ together with LATS1 and LATS2. Strikingly, every Lats1/2$^{f/f}$; YAP$^{f/f}$; TAZ$^{f/f}$; lsl-EYFP; K8CreERT2 mouse examined (n = 11) showed near-complete ablation of carcinoma development within the mammary ducts (Fig. 3b). Additionally, profiling of these mice for luminal and basal markers revealed a reversal of K14 expression in K8+ lineage-traced cells (Fig. 3c, d). Analysis of these mice confirmed the deletion of YAP and TAZ in lineage-traced cells (Supplementary Fig. 3b and c). Furthermore,

ex vivo organoid experiments performed with cells from these mice demonstrated that concomitant YAP/TAZ and LATS1/2 co-deletion also rescued phenotypes of aberrant organoid size and expression of mammary luminal and basal genes altered upon LATS1/2 deletion (Fig. 3e, f and Supplementary Fig. 3d, e). Together, these results demonstrated that YAP and TAZ are necessary to drive the proliferation and luminal-basal plasticity induced by LATS1/2 deletion in the luminal mammary epithelium.

### Luminal LATS1/2 loss promotes a basal-like transcriptional program phenocopying human basal-like breast cancer

As YAP/TAZ and Sox9 are transcriptional regulators, we hypothesized that the luminal-basal plasticity driven by LATS1/2 deletion is mediated through the control of gene expression. To assess the transcriptional signatures of LATS1/2-deleted cells, we sorted EYFP+ cells from Lats1/2$^{f/f}$; lsl-EYFP; K8CreERT2 and control lsl-EYFP; K8CreERT2 mammary glands and profiled them using RNA-sequencing. Analysis of differentially expressed genes revealed a striking upregulation of basal-associated genes in LATS1/2-null cells, and a downregulation of many luminal-associated genes (Fig. 4a and Supplementary Data 1). Gene set enrichment analysis (GSEA) revealed negative enrichment for processes of mammary development and differentiation, and positive enrichment for cell cycle processes, replicative processes, and epithelial-mesenchymal transition, among others (Fig. 4b, Supplementary Fig. 4a and Supplementary Data 2). We further compared our LATS1/2-deleted gene expression signature to previously identified transcriptomic signatures of normal mouse mammary luminal, luminal progenitor, and basal cells[42]. Gene set variation analysis (GSVA) revealed a positive enrichment for the signature of basal cells, and negative enrichment for that of luminal cells (Fig. 4c and Supplementary Fig. 4b). To test the similarity of our LATS-null signature to that of tumors derived from the inactivation of another breast tumor suppressor, Brca1, we derived upregulated and downregulated signatures for the *Brca1f/f; p53f/+; Blg-Cre* model of mouse basal-like breast cancer using data from a previous study[8]. GSEA with our LATS-null ranklist displayed a strong enrichment for upregulated genes in the *Brca1f/f; p53f/+; Blg-Cre* model, suggesting a correlation between LATS1/2 inactivation and basal-like carcinomas driven by the Brca1 mutation (Supplementary Fig. 4c).

In accordance with our finding that YAP/TAZ are necessary to promote tumorigenesis in our Lats1/2$^{f/f}$; K8CreERT2 model, we observed increased expression of many cancer-defined YAP/TAZ targets[43] in LATS1/2-null cells (Supplementary Fig. 4d). We furthermore took a set of previously identified direct transcriptional targets of YAP/TAZ[44] and performed GSVA to normal mouse mammary luminal, luminal progenitor, and basal cells as described prior. This revealed a strong enrichment for YAP/TAZ direct targets in the normal mouse mammary basal cell signature (Supplementary Fig. 4e). Additionally, GSEA comparing our LATS1/2-deleted signature to this signature of

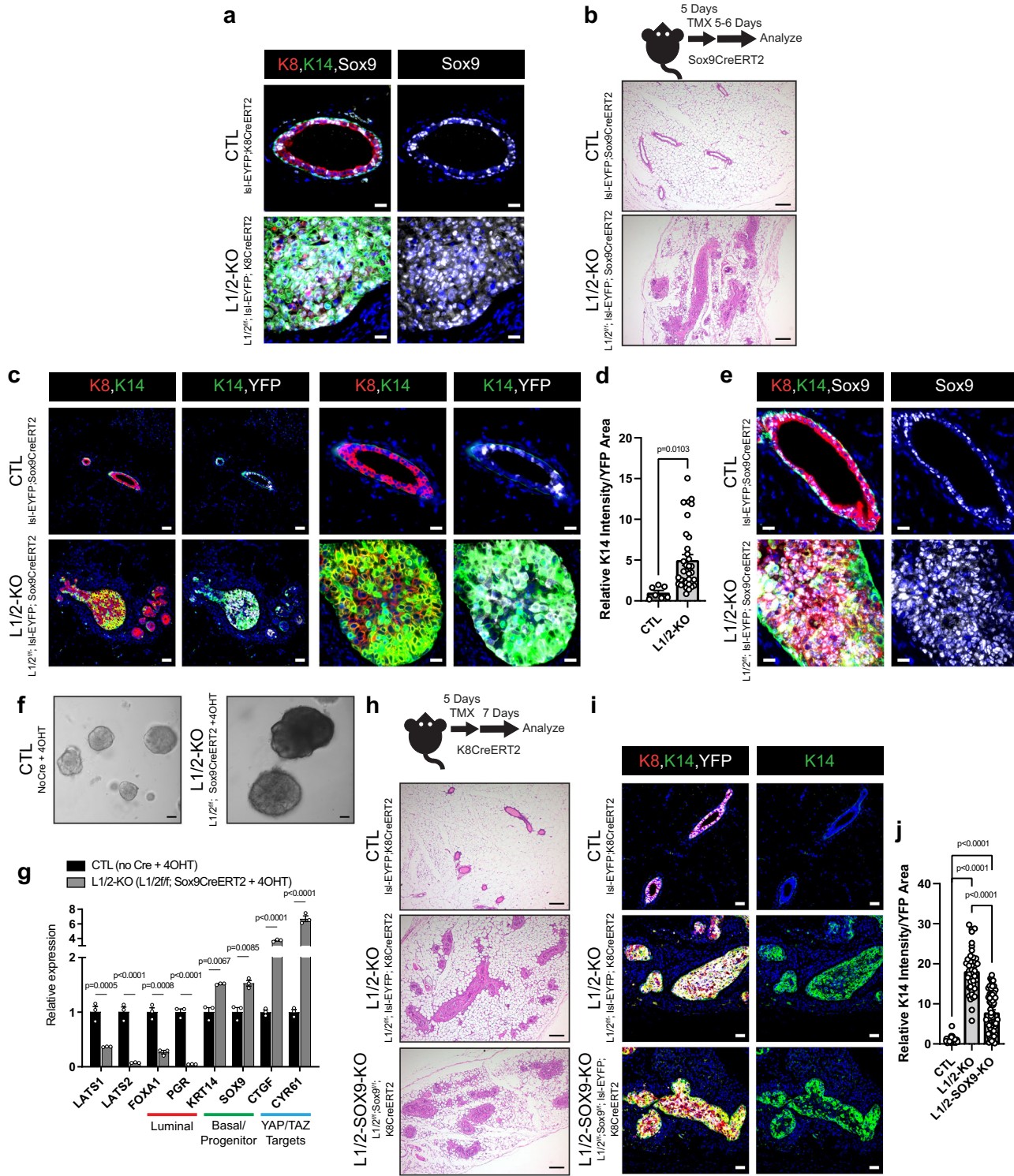

direct YAP/TAZ targets[44] revealed a significant enrichment of genes upregulated with LATS1/2 deletion in YAP/TAZ target genes (Supplementary Fig. 4f). We also performed GSEA analysis using a previously identified gene set for direct Sox9 transcriptional targets[45], which displayed a significant but less striking enrichment (Supplementary Fig. 4g). Leading edge analysis of these GSEA plots showed a divergence in YAP/TAZ and Sox9 targets (Supplementary Fig. 4h and Supplementary Data 3), suggesting that these factors promote separate gene expression programs downstream of LATS1/2 deletion. Lastly, we performed binding analysis for regulation of transcription (BART)[46] using the 500 most upregulated and downregulated genes in our

LATS1/2-deleted signature. Among the most enriched motifs in this analysis were YAP1 and several TEAD transcription factors, a family of transcription factors that are regulated by and associated with YAP/TAZ (Supplementary Data 4).

To better assess the relationship between Sox9 and the phenotypes observed following LATS1/2-deletion, we compared gene expression signatures from our LATS1/2 deletion model to those from a previous study that compared gene expression of mammary epithelial cells with high Sox9 levels (Sox9hi) and low Sox9 levels (Sox9lo) in the C3(1)-Tag basal-like breast cancer mouse model[7]. Our analysis revealed a correlation between upregulated genes in LATS1/2-deleted

**Fig. 2 | Sox9 expressing luminal cells serve as cells of origin for LATS1/2-null carcinomas. a** IF staining of K8, K14, and Sox9 in control and Lats1/2$^{f/f}$;K8CreERT2 tamoxifen-treated mammary ducts (Scale bar, 20 μM). **b** Diagram of Lats1/2$^{f/f}$;Sox9CreERT2 model used to target Sox9+ mammary epithelial cells and H&E staining of control and Lats1/2-deleted mammary glands (n = 2 for CTL) (Scale bar, 200 μM). **c** IF of K8, K14, and YFP in control and Lats1/2$^{f/f}$;lsl-EYFP;Sox9CreERT2 tamoxifen-treated mammary ducts. (Scale bar left two columns, 50 μM, Scale bar right two columns, 20 μM) (n = 2 for CTL). **d** Quantification of K14 intensity/YFP area in lsl-EYFP;Sox9CreERT2 (CTL) and Lats1/2$^{f/f}$;lsl-EYFP;Sox9CreERT2 (L1/2-KO) mice (n = 7 images from 2 CTL mice, n = 34 images from 9 L1/2-KO mice. Unpaired two-tailed t-test, Data are shown as mean ± SEM). **e** IF of K8, K14, and Sox9 in control and Lats1/2$^{f/f}$;lsl-EYFP;Sox9CreERT2 tamoxifen-treated mammary ducts (Scale bar, 20 μM) (n = 2 for CTL). **f** Morphology of organoids isolated from control and Lats1/2$^{f/f}$; Sox9CreERT2 mice and treated with 4OHT (Scale bar, 50 μM). **g** Expression of selected luminal, basal, and YAP/TAZ-induced markers in control and Lats1/2$^{f/f}$;lsl-EYFP;Sox9CreERT2 organoids treated with 4OHT (n = 3. Unpaired two-tailed t-tests. Data are shown as mean ± SEM). **h** Diagram of conditions used to co-delete LATS1/2 and Sox9 in Krt8+ luminal mammary epithelial cells along with H&E staining of control, Lats1/2$^{f/f}$;K8CreERT2, and Lats1/2$^{f/f}$;Sox9$^{f/f}$;K8CreERT2 mammary glands (Scale bar, 200 μM). **i** IF of K8, K14, and YFP in control, Lats1/2$^{f/f}$;lsl-EYFP;K8CreERT2, and Lats1/2$^{f/f}$;Sox9$^{f/f}$;lsl-EYFP;K8CreERT2 mammary ducts treated with tamoxifen (Scale bar, 50 μM). **j** Quantification of K14 intensity/YFP area in mammary ducts of lsl-EYFP;K8CreERT2 (CTL), Lats1/2$^{f/f}$;lsl-EYFP;K8CreERT2 (L1/2-KO), and Lats1/2$^{f/f}$;Sox9$^{f/f}$;lsl-EYFP;K8CreERT2 (L1/2-SOX9-KO) mice (n = 45 images from 5 CTL mice, n = 40 images from 3 L1/2-KO mice, and n = 61 images from 4 L1/2-SOX9-KO mice. One-way ANOVA with Fisher's least significant difference multiple comparisons test. Data are shown as mean ± SEM). Source data are provided as a source data file.

cells and upregulated genes in Sox9hi cells when projected onto TCGA human breast cancers using single-sample GSEA (Supplementary Fig. 4i). We also performed a similar analysis comparing Sox9 gene expression levels with the activity of genes upregulated by LATS1/2 deletion in human breast cancers, finding a strong correlation in the basal-like subtype (Supplementary Fig. 4j). These analyses together point to a transcriptional association between Sox9 levels and genes upregulated with LATS1/2 deletion in human basal-like breast cancers. RNA-sequencing on EYFP+ mammary epithelial cells isolated from Lats1/2$^{f/f}$; Sox9$^{f/f}$; lsl-EYFP; K8CreERT2 mice compared with cells from Lats1/2$^{f/f}$; lsl-EYFP; K8CreERT2 showed that deletion of Sox9 resulted in reduced expression of genes enriched for extracellular matrix (ECM) production, integrin–cell surface interactions, and neuron migration, among others, suggesting a role of Sox9 for driving these processes in the LATS1/2-null context (Supplementary Fig. 4k and Supplementary Data 2). However, Sox9 co-deletion showed no significant effects on the expression of many notable plasticity-associated luminal or basal genes compared to cells with LATS1/2 deletion alone (Supplementary Fig. 4j).

We next sought to relate our findings to human breast cancers. To assess the impact of LATS1/2 loss, we evaluated LATS1/2 copy-number alterations and LATS1/2-regulated gene signatures across the different breast cancer subtypes. Our analysis showed that loss of one or more alleles of *LATS1* or *LATS2* is widespread amongst human breast cancers, with the most notable loss in the basal subtype (Fig. 4d). This is consistent with prior reports of reduced expression of *LATS1* and *LATS2* in breast cancers[47]. Conversely, one or more alleles of *TAZ/WWTR1* are amplified across breast cancers, with the highest frequency in basal breast cancers (Fig. 4d). We extended our analysis to the genes we identified as being upregulated and downregulated in LATS1/2-deleted mammary cells, which showed differing enrichment across human breast cancer subtypes. Genes upregulated with LATS1/2 deletion were strongly enriched in the basal-like subtype relative to other subtypes, whereas the genes downregulated with LATS1/2 deletion were least enriched in the basal-like subtype (Fig. 4e). This suggests that loss of LATS1/2 activity contributes to the development of human basal-like breast carcinogenesis. Notably, we also found that human breast tumors with high activity of genes upregulated with LATS1/2 deletion in our model showed poorer survival over time, supporting the premise that LATS1/2 function as suppressors of aggressive traits in human breast cancers (Fig. 4f).

## Discussion

In this study, we examined the roles of the LATS1 and LATS2 kinases, which are central effectors of Hippo pathway signaling, in the regulation of adult mammary epithelial homeostasis in vivo. We demonstrated that deletion of LATS1/2 in mature mouse luminal cells leads to increased cell growth, luminal-basal plasticity, and initiation of basal-like mammary carcinomas dependent on the activity of the transcriptional regulators YAP and TAZ. Our study demonstrates that molecular signals mediated by these Hippo pathway effectors are required for the maintenance of luminal epithelial homeostasis and that their dysregulation drives breast cancer initiation in vivo, offering implications for treatment strategies against basal-like breast cancers, a cancer subtype with a very poor prognosis.

The cells of origin for luminal and basal subtypes of breast cancer are not fully defined, but studies suggest that basal-like cancers can arise from luminal epithelial cells[5–12]. Consistent with this premise, our observations indicate that LATS1/2 loss in K8+ luminal mammary epithelial cells is sufficient to drive basal-like tumorigenesis. Interestingly, we observed that Sox9+/ER− cells expand following LATS1/2 deletion, suggesting that loss of LATS1/2 drives the proliferation of Sox9+/ER− luminal progenitor-like cells and that these carcinomas may originate from a Sox9-expressing luminal population. Supporting this is our finding that targeted deletion of LATS1/2 in Sox9-expressing cells using a Sox9CreERT2 model resulted in basal-like tumor development like that observed using a K8CreERT2 model. However, these observations do not eliminate the possibility that LATS1/2 inactivation in ER+ luminal cells drives plasticity to a Sox9+ basal-like state as well, as the K8CreERT2 model we used efficiently targets both Sca1+ and Sca1− luminal mammary epithelial cells[48], and we observe strong repression of *Esr1* and *Pgr* in lineage-traced cells isolated from Lats1/2$^{f/f}$; lsl-EYFP; K8CreERT2 mammary carcinomas.

Prior studies have implicated LATS1/2 in the regulation of ER expression and stability, but several points arising from this prior work are worth noting in relation to our study. One recent study argues that reduced LATS1/2 activity leads to the repression of ER expression in luminal breast cancer cell lines[49], a result that mirrors our observations. However, this study also reports that inhibition of LATS1/2 represses luminal breast cancer cell growth and tumorigenesis, which contrasts with the strong tumorigenic induction we observed in vivo following LATS1/2 deletion. This difference may relate to the potential intersection of LATS1/2 deletion with other oncogenic signals existing in those cancer cell lines, which may render them unable to attain a basal-like state. Another study examined LATS1/2 in MCF10A cells and human cells enriched for basal cell traits and concluded that the presence of LATS1/2 is required to maintain the basal mammary epithelial state[50]. Our observations in vivo and in vitro oppose these conclusions, as we observe that loss of LATS1/2 strongly promotes a basal mammary epithelial state at the phenotypic and gene expression level. However, our observations do not discount the possibility that LATS1/2 plays distinct roles in mammary basal myoepithelial cells. In agreement with our observations, another study deleted LATS1 in the PyMT luminal B breast cancer mouse model and showed that the developing tumors adopt basal-like traits, including loss of ER expression and acquisition of K14[51]. This study and ours provide evidence that the LATS kinases play tumor-suppressive roles in the mammary epithelium and suggest that loss of LATS activity contributes to human breast cancer development. As this study and ours both utilize mouse models, it raises the possibility that microenvironment cues in the mammary gland

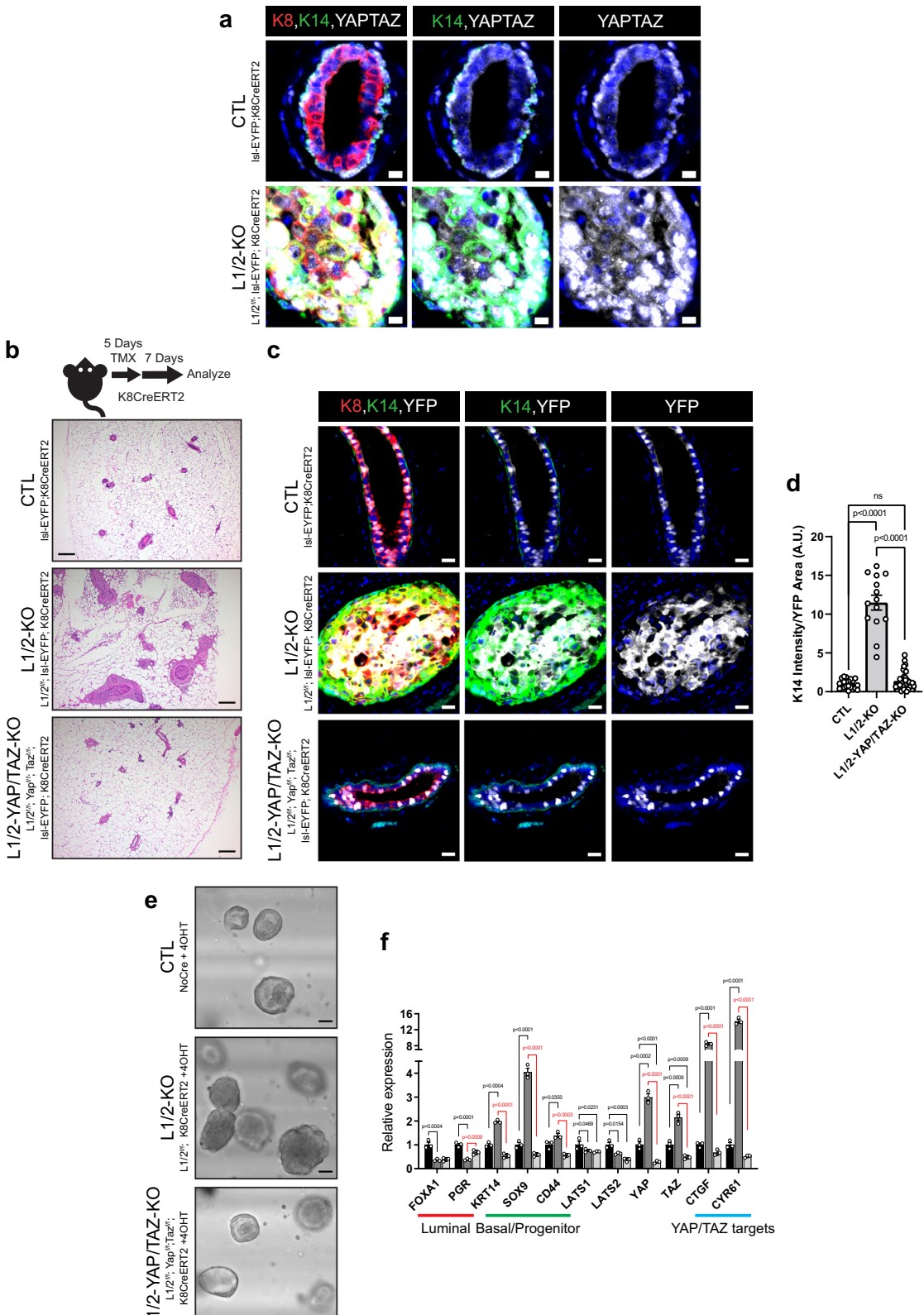

contribute to the phenotypes observed herein, and that these cues are missing in prior studies reliant on cell lines.

We found that LATS1/2 loss in the mature luminal mammary epithelium in vivo resulted in striking gene expression changes that align with transcriptional signatures observed in human basal-like breast cancers. Consistent with the aggressive traits associated with basal breast cancers, loss of LATS1/2 resulted in the development of

metastatic tumors. Intraductal viral delivery of *Ad-K8-nlsCre* to Lats1/2-floxed mammary ducts, which offers localized targeting of luminal epithelial cells within the mammary ductal tree, leads to the development of tumors that show luminal-basal plasticity and features of an epithelial–mesenchymal transition. Our genetic experiments further demonstrated that YAP/TAZ are essential factors downstream of LATS1/2, as co-deletion of YAP/TAZ rescued the tumorigenic

**Fig. 3 | Mammary carcinomas driven by LATS1/2 loss are dependent on YAP and TAZ. a** IF of K8, K14, and YAP/TAZ in control and Lats1/2$^{f/f}$;lsl-EYFP;K8CreERT2 mammary ducts (Scale bar, 10 µM). **b** Diagram of conditions used to co-delete LATS1/2, YAP, and TAZ in K8$^+$ luminal mammary epithelial cells and H&E staining of control, Lats1/2$^{f/f}$;K8CreERT2, and Lats1/2$^{f/f}$;Yap$^{f/f}$;Taz$^{f/f}$;lsl-EYFP;K8CreERT2 mammary glands treated with tamoxifen (Scale bar, 200 µM). **c** IF of K8, K14, and YFP in control, Lats1/2$^{f/f}$;lsl-EYFP;K8CreERT2, and Lats1/2$^{f/f}$;Yap$^{f/f}$;Taz$^{f/f}$;lsl-EYFP;K8CreERT2 mammary ducts (Scale bar, 20 µM). **d** Quantification of K14 intensity/YFP area in mammary ducts of lsl-EYFP;K8CreERT2 (CTL), Lats1/2$^{f/f}$;lsl-EYFP;K8CreERT2 (L1/2-KO), and Lats1/2$^{f/f}$;Yap$^{f/f}$;Taz$^{f/f}$;lsl-EYFP;K8CreERT2 (L1/2-YAP/TAZ-KO) mice ($n = 20$

images from 3 CTL mice, $n = 14$ images from 3 L1/2-KO mice, $n = 37$ images from 6 L1/2-YAP/TAZ-KO mice. One-way ANOVA with Fisher's least significant difference multiple comparisons test. Data are shown as mean ± SEM). **e** Morphology of organoids cultured from control, Lats1/2$^{f/f}$;K8CreERT2, and Lats1/2$^{f/f}$;Yap$^{f/f}$;Taz$^{f/f}$; K8CreERT2 mice treated with 4OHT (Scale bar, 65 µM). **f** Expression of selected luminal, basal/stem-cell, and YAP/TAZ-induced markers in organoids cultured from control, Lats1/2$^{f/f}$;K8CreERT2, and Lats1/2$^{f/f}$;Yap$^{f/f}$;Taz$^{f/f}$;K8CreERT2 mice treated with 4OHT ($n = 3$. One-way ANOVA with Fisher's least significant difference multiple comparisons test. Data are shown as mean ± SEM). Source data are provided as a source data file.

---

phenotypes arising from LATS1/2 deletion. Interestingly, prior work and ours have also suggested a correlation between Sox9 levels and the phenotypes observed in LATS1/2-deleted mammary epithelial cells. Previous ATAC-seq data from mouse basal-like mammary carcinoma cells with high Sox9 levels[7] demonstrated open chromatin enriched for the TEAD transcription factor motif, suggesting potential activation of both YAP/TAZ and Sox9 signaling in these cells. Further, we found that gene expression signatures from basal-like breast carcinoma cells with high Sox9 levels correlate with those from LATS1/2-deleted luminal mammary epithelial cells. Co-deletion of Sox9 together with Lats1/2 did not overtly rescue the observed tumorigenic phenotype we observed with LATS1/2 deletion alone but did reduce the induction of K14 protein in the targeted luminal cells. However, the effects of Sox9 deletion in our model were modest when compared to YAP/TAZ deletion, indicating that YAP/TAZ are more central transcriptional effectors downstream of LATS1/2 deletion in the mammary epithelium.

Our study identifies YAP/TAZ as the primary signaling effectors downstream of LATS1/2 that drive luminal-basal plasticity in basal-like breast cancer, suggesting that abrogation of YAP/TAZ activity may be beneficial for the treatment of basal-like breast cancers. TAZ in particular has been highlighted by prior studies as a crucial factor that promotes breast cancer stem cell properties and oncogenesis, with notably elevated levels in late-stage human breast cancers[16]. Ectopic TAZ expression in luminal mammary epithelial cells has also been shown to drive basal features[15], resulting in phenotypes that resemble those we have observed with LATS1/2 deletion. TAZ and YAP are key effectors of mechanotransduction[52] and intersect with mechanically sensitive growth factor-induced signaling pathways[53,54]. Thus, an altered mammary microenvironment likely contributes to the dysregulation of LATS1/2 and YAP/TAZ activity in breast tumors. Opportunities to target YAP and/or TAZ activity pharmacologically have hinged primarily on disrupting the association of YAP/TAZ with the TEAD transcription factors or identifying and targeting upstream regulators or downstream targets of YAP/TAZ[23]. Further studies will be necessary to test how basal-like breast cancers may respond to such targeted therapies.

In summary, our results demonstrate that conditional loss of LATS1/2 in the mature mouse luminal mammary epithelium results in the development of mammary carcinomas with basal-like traits. These carcinomas develop in a YAP/TAZ-dependent manner and resemble human basal-like breast cancers. This study, therefore, offers insight and clarification into the functions of Hippo pathway signaling in the mammary epithelium and basal-like breast cancer initiation.

## Methods
### Mice
All animal experiments were conducted according to protocols approved by the Institutional Animal Care and Use Committee (IACUC) at Boston University (Protocol # PROTO201800389). All mice used in this study are listed in Supplementary Table 1. Nulliparous mice were used for all experiments and were aged to adulthood (over 12 weeks) prior to all experiments. Mice were housed under conditions of 68–79 °F, 30–70% humidity, and 12 h light/dark cycles. The maximum tumor size permitted by Boston University is 20 mm in diameter, and

no tumor was measured to exceed this size. For experiments with Tamoxifen treatment, all mice received 2 mg of Tamoxifen (Sigma, T5648) diluted in corn oil (Sigma, C8267) intraperitoneally on 5 consecutive days. Endpoints for Lats1/2$^{f/f}$; lsl-EYFP; K8CreERT2, Lats1/2$^{f/f}$; Sox9$^{f/f}$; lsl-EYFP; K8CreERT2, and Lats1/2$^{f/f}$; YAP$^{f/f}$; TAZ$^{f/f}$; lsl-EYFP; K8CreERT2 utilized in this study ranged from 1 to 10 days after the last Tamoxifen dose. Endpoints for Lats1/2$^{f/f}$; lsl-EYFP; Sox9CreERT2 mice ranged from 4 to 13 days after the last Tamoxifen dose, and these mice were generally collected earlier due to more rapid deterioration of animal health.

### Immunofluorescence and immunohistochemistry
Tissues were excised from animals and fixed in 4% paraformaldehyde (PFA) (Electron Microscopy Sciences, # 15710) at room temperature overnight. They were then processed for paraffin embedding. Staining was preceded by dewaxing and ethanol dehydration, followed by antigen retrieval using an unmasking solution (Vector Laboratories, H3300). Antigen retrieval was performed via either microwave or decloaking chamber (Biocare Medical, DC2012). Hematoxylin (Sigma, catalog no. MHS16) and Eosin (Sigma, catalog no. HT110116) staining were performed according to the manufacturer's instructions. For immunofluorescence (IF) staining, samples were blocked with 5% donkey serum (EMD Millipore, 50-588-37) in TBS-T, followed by incubation in primary antibodies overnight at 4 °C. For unconjugated mouse antibody staining, an additional blocking step was included using Rodent Block M (Biocare Medical, RBM961). After overnight incubation, samples were washed in TBS-T and incubated in a secondary antibody for one hour. Information regarding primary and secondary antibodies is supplied in Supplementary Table 2. All slides were mounted using Prolong Gold antifade reagent (Invitrogen, P36930, or P36931). Nuclei were stained using DAPI or 1 µg/mL Hoescht. Images were captured using a Zeiss Axio Observer.Z1 microscope with Zeiss ZEN 3.3 Blue software, a Zeiss LSM 700 microscope with ZEN 2.3 SP1 FP3 Black software, or a Zeiss Axio Scan.Z1 microscope with Zeiss ZEN 3.1 Blue software. Histology for Lats1/2$^{f/f}$; lsl-EYFP; K8CreERT2, Lats1/2$^{f/f}$; Sox9$^{f/f}$; lsl-EYFP; K8CreERT2, and Lats1/2$^{f/f}$; YAP$^{f/f}$; TAZ$^{f/f}$; lsl-EYFP; K8CreERT2 mice displayed in figures was performed on mice collected 7 days after the last Tamoxifen dose, except where noted. Histology for Lats1/2$^{f/f}$; lsl-EYFP; Sox9CreERT2 mice; and lsl-EYFP; Sox9CreERT2 mice displayed in figures was performed on mice collected 5–6 days after the last Tamoxifen dose and 7 days after the last Tamoxifen dose, respectively.

### Mammary organoid cultures
Mammary organoids were obtained and cultured using a previous protocol as a basis[32]. Briefly, immediately upon excision, the third, fourth, and fifth mammary glands were chopped with a McIlwain tissue chopper (Ted Pella, Inc.) prior to being digested with a collagenase solution. The collagenase solution contained 2 mg/mL collagenase (Roche, 11088793001 and Worthington, LS004196), 0.5 units/mL dispase (Corning, 354235), 10 µg/mL DNAse (Roche, 4716728001), and 1× Penicillin–Streptomycin (Corning, 30-002-CI) diluted in DMEM (Corning, 10-013-CV). Red blood cells were lysed using ACK Lysing buffer (Gibco, A1049201). Two methods were used to eliminate

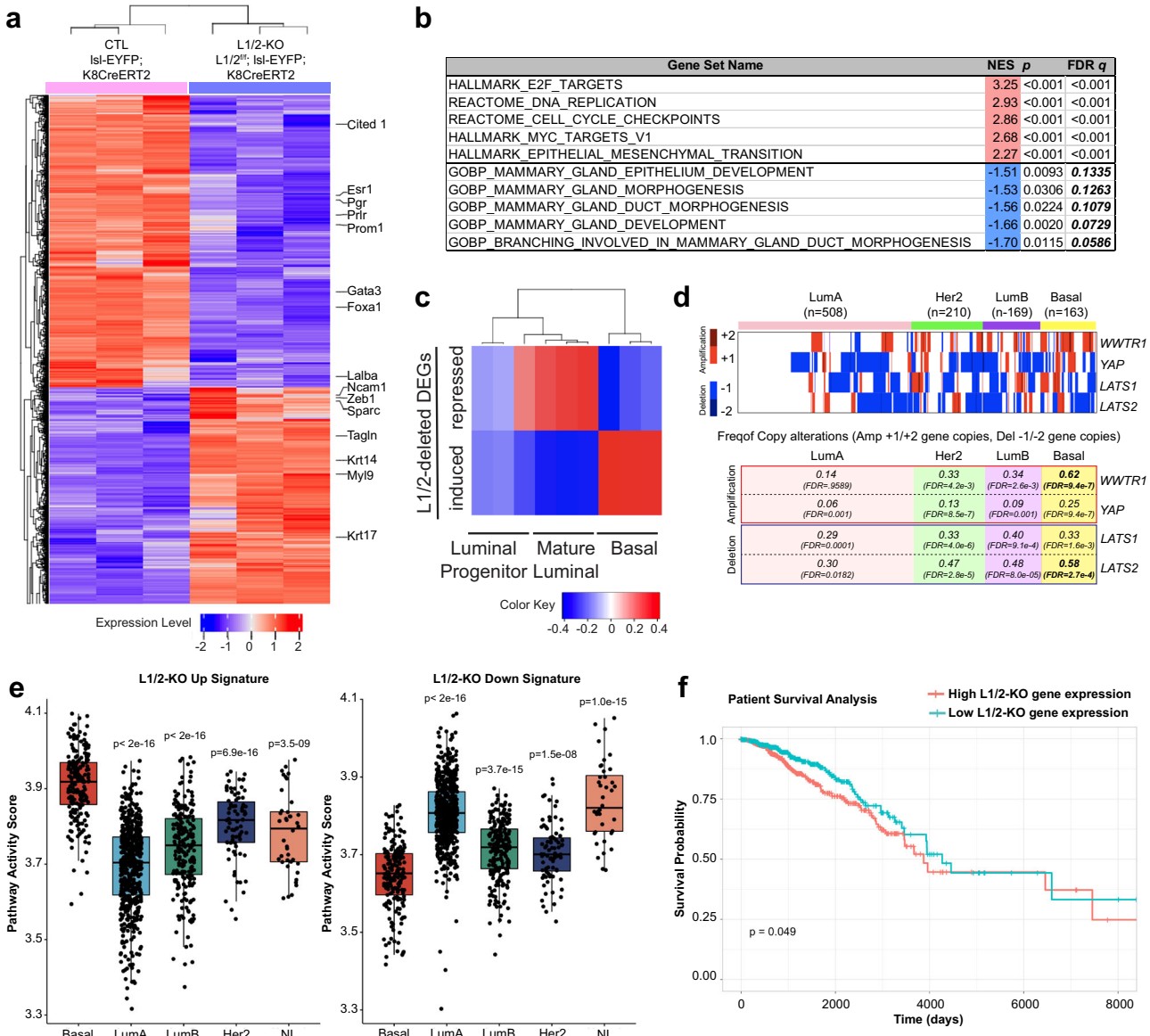

**Fig. 4 | Luminal LATS1/2 loss promotes a basal-like transcriptional program phenocopying human basal-like breast cancer. a** Heatmap of gene expression changes in EYFP+ mammary cells sorted from Lats1/2$^{f/f}$;lsl-EYFP;K8CreERT2 mice compared to lsl-EYFP;K8CreERT2 control mice as analyzed by RNA-sequencing (*n* = 3 per condition, 2-fold change and FDR < 0.05 cutoff). **b** Selected GSEA-generated genesets enriched in the upregulated (red) and downregulated (blue) signatures of LATS1/2-null cells compared to control. **c** GSVA analysis comparing the upregulated and downregulated signatures of LATS1/2-null cells to those of normal mammary luminal mature, luminal progenitor, and basal cells (GSE63310). **d** Analysis of copy number alterations of Hippo pathway-related factors in human breast cancers. Heatmap representing one or two allele deletions or amplifications are shown on top and the frequency of these alterations in the different breast cancer subtypes are shown at the bottom. **e** Activity levels of the LATS-up/

downregulated genes in TCGA-BRCA samples across the Basal, HER2+, Luminal A, Luminal B, and Normal-like (NL) subtypes (*n* = 189(Basal), 564(LumA), 215(LumB), 82(Her2), 40(NL). Two-sided *t*-test. *p*-values represent comparisons to the Basal group. No adjustment for multiple comparisons. Each box plot draws the center line at the median value, the upper and lower box boundaries at the first and third quartiles (25th and 75th percentiles), and the whiskers at ±1.5 × the interquartile range). **f** Breast cancer patients with survival information (*n* = 1102) were divided into two groups based on high and low gene expression following LATS1/2 deletion based on the median LATS1/2 gene expression scores (*n* = 551 per group), and these gene expression signatures were compared for their overall survival outcomes. The statistical significance of survival difference between groups was determined using a log-rank test.

stromal cells from the organoid suspension: differential centrifugation using 5–10 rapid spins per sample, and differential adhesion plating by incubating organoids and residual stromal cells in tissue culture-treated plates (Falcon, 08-772E) for 1–2 h. Following fibroblast depletion, organoids were resuspended in Matrigel (Corning, 356237) and seeded on round coverslips (Fisherbrand, 12-545-80 and 12-545-81) in 24-well plates. Organoids were grown in media containing DMEM/F12 (Gibco, 21331020), 1× Insulin-Transferrin-Selenium (Corning, 51500056), 1x Penicillin–Streptomycin (Corning, 30-002-CI), and

2.5 nM EGF (Corning, C354001). For adenovirus infection of organoids, infection was performed immediately prior to seeding in Matrigel. Infection was performed by resuspended organoids in DMEM (Corning, 10-013-CV) (no additives) and incubating in adenovirus at a concentration of 1000 pfu/organoid for 2 h. Afterward, organoids were centrifuged, washed with DMEM, resuspended in Matrigel, and seeded in 24-well plates as described above. For 4-hydroxy-tamoxifen (Sigma-Aldrich, H7904) treatment of organoids, treatment began 7 days after seeding. 4-hydroxy-tamoxifen treatment was performed at 0.1 μM.

Imaging of organoids was performed using a Zeiss Axio Observer.Z1 with Zeiss ZEN 3.3 Blue software and a Zeiss SteREO Discovery.V12 microscope using Zeiss ZEN 2 Blue software.

## Flow cytometry and FACS

For flow cytometry experiments, mammary glands were excised and digested as described for organoid experiments. Following digestion at 37 °C, samples were dissociated into single cells using TrypLE (Gibco, 12604-021), quenched with DMEM/10% FBS, and washed with PBS. Cells were then put through a 40 μM (Fisher, 22-363-547) or 70 μM strainer (Corning, 352350). For mammary gland profiling, live/dead staining was performed using a near-IR dead cell stain (Invitrogen, L34976), followed by blocking using Fc block (BD Biosciences, 553142) in BD stain buffer (BD, 554656). Samples were then incubated in primary antibodies at the specified concentrations for 30 min at 4 °C, washed twice, and resuspended in PBS. Compensation was performed using UltraComp eBeads Compensation Beads (Invitrogen). Flow cytometry was performed using a BD LSRII flow cytometer using FACS Diva 6.2.1 (BD Biosciences) and analysis was performed using FlowJo v10.7.1 (BD Biosciences). All antibodies used for flow cytometry are outlined in Supplementary Table 2. For FACS experiments, live/dead staining was performed using Calcein blue (Invitrogen, C1429) at a concentration of 1 μM. Cell sorting was performed on a Moflo Astrios flow cytometer (Beckman Coulter).

## Intraductal adenovirus injections

Ad5mK8-nlsCre (VVC-Li-535; University of Iowa Viral Vector Core) was diluted in DMEM with CaCl2 (10 mM) and introduced into the fourth mammary ducts at a titer of ~$1.92 \times 10^7$ pfu/duct using 100 μL Nanofil syringes (World Precision Instruments) fitted with 35 G Nanofil needles (World Precision Instruments). Live animal and organ imaging was performed using an IVIS Spectrum imaging system (Perkin Elmer). Imaging of lung metastases was performed on a Zeiss Axio Observer.Z1 microscope with Zeiss ZEN 3.3 Blue software.

## RNA Isolation and RT-qPCR

RNA isolation from mammary organoid cultures was performed using the RNeasy Mini Kit (Qiagen, 74106) following the manufacturer's instructions. Matrigel cultures were lysed directly with RLT buffer. Reverse transcription was performed using the iScript cDNA Synthesis Kit (BioRad, 1708891) following the manufacturer's instructions. RT-qPCR analysis was performed on the ViiA7 Real-Time PCR System (Applied Biosystems) with QuantStudio real-time PCR Software v1.6.1 (Applied Biosystems) using SYBR protocols with Fast SYBR Green Master Mix (Applied Biosystems, 4385618). All primers used for RT-qPCR in this study are outlined in Supplementary Table 3.

## RNA-sequencing and global gene expression analyses

FACS-sorted cells were lysed using Trizol LS, followed by RNA isolation using the RNEasy Mini Kit (Qiagen, 74106). Library preparation for RNAseq was performed using the Illumina TruSeq RNA Sample Preparation Kit and RNA sequencing was performed on a NextSeq 2000. FASTQ files were aligned to mouse genome build mm10 using STAR (version 2.6.0c). EnsemblGene-level counts for non-mitochondrial genes were generated using featureCounts (Subread package, version 1.6.2) and Ensembl annotation builds 100 (uniquely aligned proper pairs, same strand). SAMtools (version 1.9) was used to count reads, FASTQ quality was assessed using FastQC (version 0.11.7), and alignment quality was assessed using RSeQC (version 3.0.0). Variance-stabilizing transformation (VST) was accomplished using the variance-stabilizing transformation function and differential expression was assessed in the DESeq2 R package (version 1.23.10). Correction for multiple hypothesis testing was accomplished using the Benjamini–Hochberg false discovery

rate (FDR). Human homologs of mouse genes were identified using HomoloGene (version 68). Differential gene expression signatures (DEGs) were calculated with a threshold of absolute log2FC > 1 (up and down respectively) and FDR ≤ 0.05. DEGs heatmap visualization between KO/WT samples was performed with the R package ComplexHeatmap (v2.6.2). RNA sequencing data have been deposited to NCBI GEO (GSE196555).

Gene set enrichment analysis (GSEA) (version 2.2.1) was used to identify biological terms, pathways, and processes that are coordinately up- or down-regulated within each pairwise comparison. The Entrez Gene identifiers of the human homologs of all genes in the Ensembl Gene annotation were ranked by the Wald statistic computed for each pairwise comparison. Ensembl Genes matching multiple mouse Entrez Gene identifiers, and mouse genes with multiple human homologs (or vice versa), were excluded prior to ranking so that the ranked list represents only those human Entrez Gene IDs that match exactly one mouse Ensembl Gene. Each ranked list was then used to perform pre-ranked GSEA analyses (default parameters with random seed 1234) using the Entrez Gene versions of the Hallmark, Biocarta, KEGG, PID, Reactome, WikiPathways, Gene Ontology (GO), and transcription factor and microRNA motif gene set obtained from the Molecular Signatures Database (MSigDB), version 7.4. Nominal *p*-values for GSEA analyses were generated using a one-sided permutation test, and FDR *q*-values were calculated from nominal *p*-values using the Benjamini–Hochberg procedure. Signatures of *Brca1f/f; p53f/+; Blg-Cre* tumors were generated by comparing the expression profiles of 12 *Brca1f/f; p53f/+; Blg-Cre* tumors to three normal LumER-samples using data obtained from ArrayExpress (E-TABM-683 and E-TABM-684), using a cutoff of logFC > 2 and FDR < 0.01. The signatures generated included 1697 upregulated and 859 downregulated genes.

## Gene expression signature projection into The Cancer Genome Atlas Breast Cancer (TCGA-BRCA) subtypes

Transcriptional activity of the signatures was projected in TCGA-BRCA primary tumors (*n* = 1102) using a single sample GSEA (ssGSEA) method from the R package GSVA (v1.34.0). TCGA-BRCA RNA-Seq count matrix (STAR 2-Pass/HTSeq-Counts) was downloaded through gdc-client from the Genomic Data Commons. Counts underwent a variance-stabilizing transformation using DESeq2 followed by a log transformation. Signature activation was compared between PAM50 (mRNA) molecular subtypes[55]. 1090 samples had subtype classifications (LumA = 564, LumB = 215, Basal = 189, Her2 = 82, NL = 40).

## Correlation analysis of LATS1/2-regulated genes with Sox9hi gene signature

Microarray expression data for FACS-sorted Sox9 GFP-high/low cells (*n* = 6, 3 per group) was obtained from GEO (GSE135885, GSE135891). Differential expression between high and low groups was performed with Limma (FDR < 0.05 and log2FC > ±1), producing Sox9-high upregulated and downregulates signatures of 430 and 480 genes, respectively. Transcriptional activity of the signatures was projected in TCGA-BRCA in addition to LATS signatures as previously described. The transcription activity of signatures was compared across TCGA-BRCA samples with Pearson correlation.

## GSVA analysis comparing LATS1/2-regulated genes to mammary epithelial cell populations

RNA-seq counts-level data were obtained from GEO (GSE63310) for 9 samples, 3 from each group, including basal cells, luminal progenitor-enriched cells, and mature luminal-enriched cells. Counts were normalized by Trimmed Mean of *M*-values (TMM). Pairwise differential expression analysis was conducted between groups using Limma-Voom (v3.46). Genes with FDR ≤ 0.05 and log2FC ≥ ±1 across pairwise experiments were pooled (1692 genes) and used to generate a gene expression heatmap for all 9 samples.

## Copy number analysis

For allele copy number analysis, there were 1080 breast cancer tumor samples used in the original GISTIC analysis using GISTIC version 2.0.22 (Firehose task version:140). Gene copy number alteration levels were examined from 1050 breast tumor samples with matching gene expression data. Copy number alterations were determined using gene-based copy number levels ([−2, −1, 0, 1, 2]) for each tumor sample. Copy number levels of [1,2] and [−1,−2] were labeled as amplifications and deletions, respectively. Of note, cBioPortal uses a stricter cutoff for binarization, so deletions correspond to [−2] and amplifications correspond to [+2]. For each gene-level alteration of interest, we tested for the significance of the increased frequency of copy number alterations compared to the expected background frequency using $binomial.test(x, n, p)$, where $x$ is the number of observed amplification (or deletion) events, $n$ is the total sample size, and $p$ is the background probability of alteration, estimated using the fraction of total amplification or deletion events. This analysis was repeated for each breast cancer subtype.

## Patient survival analysis

Transcriptional activity signatures projected in TCGA-BRCA primary tumors (by ssGSEA) were used to stratify breast cancer patients with survival information ($n = 1102$) into two groups based on the high and low activity of the expression signatures following LATS1/2 deletion. Total activity was computed as the LATS-upregulated activity score minus the LATS-downregulated activity score. Samples were split based on the median ($n = 551$ per group).

## Image quantification

All image quantification was performed using CellProfiler 4.2.1[56]. For quantification of immunofluorescence images, individual images of mammary ducts were thresholded on YFP, then masked on the thresholded YFP image using MaskImage. The total intensity of the K14 channel was then measured within the masked YFP+ cells using MeasureImageIntensity. The masked YFP area per image was measured using MeasureImageAreaOccupied, and K14 intensity/YFP area per image was calculated by dividing the total intensity of K14 within the masked image by the total masked YFP area per image. These values were then normalized to the average values of images from respective control mice within each experiment to obtain relative K14 intensity/YFP area values. For the quantification of organoids, the edges of organoids were enhanced using EnhanceEdges, followed by the designation of organoids as primary objects using IdentifyPrimaryObjects. The area of individual objects was then measured using MeasureObjectSizeShape, followed by the calculation of the average object area per well as displayed.

## Statistics and reproducibility

All immunofluorescence stainings, hematoxylin and eosin stainings, brightfield organoid imaging, and organoid size quantifications shown in the figures were performed across at least three independent experiments with similar results, except when noted in the legends. Statistical analyses were performed in GraphPad Prism 9. All $T$-tests performed were performed without correction for multiple comparisons. Statistical analyses of RT-qPCR data were performed on log10-transformed fold change values.

## Reporting summary

Further information on research design is available in the Nature Portfolio Reporting Summary linked to this article.

## Data availability

TCGA datasets used in the study are publicly available and can be obtained from https://portal.gdc.cancer.gov. Gene expression data used in the study, including new data that was generated, are publicly available on the GEO repository: GSE196555, GSE135885, GSE135891, and GSE63310. The authors declare that all other data supporting this study are either available within the article, supplementary information, and source data. Source data are provided with this paper.

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

## Acknowledgements

We would like to acknowledge support from the Boston University Flow Cytometry Core and Microarray and Sequencing Resource Core Facility, with particular thanks to Adam Gower. We thank Jeffrey Wrana for sharing the Yap-loxP/loxP and Wwtr1-loxP/loxP mice. This work was supported by funds from the Dahod Grant Program for breast cancer research at Boston University. X.V. was funded by NIH/NHLBI R01HL124392, NIH/NIDCR R01DE030350, and by an ACS Research Scholar Grant (RSG-17-138-01-CSM). J.G.K. was funded by NIH/NCI grant F31CA232683. S.M. was in part funded by Find the Cause Breast Cancer Foundation and NIH/NIDCR R01DE031831. E.S. was funded by a Susan G. Komen Foundation Graduate Training in Breast Cancer Disparities Research Grant. R.L.J. was funded by CPRIT awards RP200240 and RP180530. A.F. and B.N. were funded by Moorman–Simon Fellowships in Computational Biomedicine.

## Author contributions

J.G.K., A.M.T.-L., G.B.P., and E.S. performed experiments. A.F., B.N., and A.M. performed computational analyses of data. N.C. assisted with data analysis. R.L.J. supplied the Lats1$^{f/f}$ and Lats2$^{f/f}$ mice. M.E.L., J.E.B., S.M., and X.V. supervised the study. J.G.K. and X.V. wrote the manuscript.

## Competing interests

The authors declare no competing interests.
