## [Peer Review File · Nature Communications]

Inactivation of LATS1/2 drives luminal-basal plasticity to initiate basal-like mammary carcinomasREVIEWER COMMENTS

Reviewer #1 (Remarks to the Author):

Kern et al have shown the contribution of LATS1/2 in the context of maintaining luminal cell in mammary epithelium and loss of LATS1/2 results in development of basal like breast cancers by utilizing mouse models. The authors have also shown the contribution of SOX9 in promoting metastasis of the tumors.

The interesting finding is the link between SOX9 and Hippo pathway effectors, YAP1 and WWTR1.

Major comments.

1. The authors could try to characterize the mechanisms involving SOX9 and YAP1 (and/or WWTR1) which will strengthen the story, even though the role of SOX9 in luminal to basal plasticity is dispensable. This would probably shed light on the mechanisms and identify therapeutic targets to treat the basal like breast cancers.

2. Since YAP1 and WWTR1 are transcriptional regulators, it would have been helpful to identify transcriptional targets that govern the luminal to basal plasticity by performing chromatin immunoprecipitation experiments including SOX9 as well.

3. It was previously suggested that luminal progenitor cells with BRCA1 mutation, could be the cell of origin for basal like breast cancers (also referenced by the authors). How does YAP1/WWTR1 correlate to BRCA1? or LATS1/2 deletion in mature luminal correlate to Basal like breast cancers that arise as a result of BRCA1 mutation. Any suggested mechanisms? based on their gene expression analysis and GSEA analysis.

4. Hippo pathway was previously shown to contribute to metastasis. Mechanistically, how does SOX9 and YAP1/WWTR1 function together in this context?

The authors have made an attempt and utilized GSEA analysis to compare LATS up-regulated signature to SOX9 up-regulated signature in S4E. The selected genes as shown in heatmap from S4C and S4F has distinct genes enriched. Detailed analysis of the intersection would increase the strength of the manuscript.

Reviewer #2 (Remarks to the Author):

The paper "Inactivation of LATS1/2 drives luminal-basal plasticity to initiate basal-like mammary carcinomas" written by Kern et al reported that LATS1/2 knockout in mature mammary luminal epithelial cells drove the development of basal-like breast cancers. Notably, gene expression analyses of LATS1/2-deleted mammary epithelial cells revealed a transcriptional program that associates with human basal-like breast cancers. In general, the experiments and data analysis were conducted to a high standard and the findings are compelling. Figures are clearly laid out and the manuscript is well written. Since this is an area of controversy this work adds more clarity to the role of Lats (and Hippo signaling) in basal breast cancer:

Comments:

1. In Fig. 1D and 2C, the IF data displayed K8/K14 and K14/YFP, respectively. The authors should consider showing K8 and YFP colocalization to indicate that the cells expanding in mammary ducts are YFP+ cells. This would be a more impactful way to show the data.

2. In Figure 1e and sFig1b, please clarify if the reduction in Sca1-expressing cells resulted from low EYFP+ cells?

3. Is there any relationship between YAP or YAP targets and basal-like cell markers (basal cell state)?

4. Please comment regarding SOX9 expression in human basal-like breast cancers? Is there any overlap with low Lats1/2 or high YAP activity?

5. Are K8 and SOX9 co-expressed in same luminal cells? Is there any difference between Lats1/2f/f; Isl-EYFP; K8CreERT2 and Lats1/2f/f; Isl-EYFP; Sox9CreERT2 mice (such as cell proliferation speed and K8/K14+ cells)?

Minor:

1. "We next sought to relay our findings to human breast cancers." Should be relate

Reviewer #3 (Remarks to the Author):

This paper is interesting and relevant to mammary cancer biology and its intersection with Hippo signaling. For the most part it is well written and conclusions are appropriate to the data. Several statements would benefit from more information, clarification and/or quantification. In addition, while the mammary tumor experiments with the various TG mouse models are undeniably elegant, there should be more restraint shown when conclusions are made directly relevant to breast cancers. For instance, loss of LATS1/2 in normal finite human mammary epithelial cells (as in not MCF10A or 184A1) is not sufficient to cause transformation, whereas in these mouse experiments loss of LATS1/2 results in mammary tumors. There are clearly barriers to progression that are missing in mice, and the interpretation should reflect those evolutionary differences.

Figure 1, Supp. Figure 1 and accompanied text-

-The reader may benefit if the text noted that phosphorylated LATS1/2 are generally thought to inactivate YAP/TAZ function.

-General lack of quantification for IF images. How many mice and sections were looked at?

Recommend quantification of specific IF images to make key points.

-Appreciate the rigor of using 2 ex vivo organoid culture models to test LATS1/2 function in other contexts. Also appreciate addressing tamoxifen treatments as a confounding factor.

-Supp. Figure 1H

Appreciate the controls testing if Cre is only active in K8 luminal cells. It is a bit difficult to tell. This would be enhanced by including an image that is zoomed in, or perhaps through quantification.

-Figure 1J- Provide a more zoomed in picture. Or some sort of quantification. It is a bit difficult to find K8/K14/tdtomato cells.

-Please include important information: How many mice were used in each experiment? How many control/ko mice developed tumors? How many control/ko mice developed lesions?

Figure 2, Supp. Figure 2 and accompanied text-

-Please include important information: How many mice were used in each experiment? How many control/ko mice developed tumors? How many control/ko mice developed lesions?

-Why do the authors think the cancers occurred more rapidly when knockout was introduced in Sox9+ cells? – generally curious about their interpretation

-Fig 2C, why is there a lack of K14 staining in zoomed in control images?

-General lack of quantification for IF images. How many mice and sections were looked at?

Recommend quantification of specific IF images to make key points.

Figure 3, Supp. Figure 3 and accompanied text-

-Figure 3A- YAP/TAZ expression in control myoepithelial cells is not convincing.

-Figure 3 would benefit from quantification of IF images as it would help to accurately assess statements in text.

-How many control/ko mice had DCIS? How many mice and slices were looked at?

- Supp. Figure 3b-3c, It would be nice to have a zoomed in image of just YFP and just YAP in the same image to assess what is said in the text "reliable deletion of YAP and TAZ".

-Quantify size of organoids. It would be nice to quantify the size of the organoids to assess for statistically significant morphological changes between conditions.

Regarding Figure 4, Supp Figure 4 and accompanied text-

-Reasonable and supportive of findings found in previous figures.

Response to Reviewers

We thank the reviewers for their positive assessment of our study and for the helpful suggestions that have led to an improved manuscript. Below we provide a point-by-point response to the reviewers' comments with our responses in blue.

Reviewer #1

Kern et al have shown the contribution of LATS1/2 in the context of maintaining luminal cell in mammary epithelium and loss of LATS1/2 results in development of basal like breast cancers by utilizing mouse models. The authors have also shown the contribution of SOX9 in promoting metastasis of the tumors. The interesting finding is the link between SOX9 and Hippo pathway effectors, YAP1 and WWTR1.

Response:

We thank the reviewer for their suggestions to improve our manuscript and have done our best to address all the comments raised.

We agree that identifying the relative contributions of YAP/TAZ and Sox9 in promoting mammary carcinomas is an important goal. We also would like to clarify that we do not functionally demonstrate that Sox9 promotes metastasis and show that while Sox9 partially contributes to the phenotypes observed following LATS1/2-deletion, YAP/TAZ are more central to these phenotypes.

Major comments.

1. The authors could try to characterize the mechanisms involving SOX9 and YAP1 (and/or WWTR1) which will strengthen the story, even though the role of SOX9 in luminal to basal plasticity is dispensable. This would probably shed light on the mechanisms and identify therapeutic targets to treat the basal like breast cancers.

Response:

We thank the reviewer for raising an interesting point, motivating us to perform a more careful analysis of Sox9 in relation to YAP/TAZ and LATS1/2 deletion in the revised manuscript.

This characterization included temporally tracking Sox9 positive cells in mammary ducts of LATS1/2^{f/f}; Isl-EYFP; K8CreERT2 mice early (1, 2, 3, 4, 5, 6 days after the last Tamoxifen dose) following LATS1/2 deletion, where we observed that early arising proliferating cells that transition to a K14 basal-like state are Sox9 positive. These results are now displayed in Supplementary Figure 2B. This is in concordance with our observations that tamoxifen treatment of LATS1/2^{f/f}; Isl-EYFP; Sox9CreERT2 mice (Fig 2C-2E), which deletes LATS1/2 in Sox9-positive mammary epithelial cells, promotes similar basal-like overgrowth.

We also better quantified the roles of Sox9 in the phenotypes observed following LATS1/2-deletion and mapped Sox9-regulated target genes in this context. Our quantitation of LATS1/2^{f/f}; Sox9^{f/f}; Isl-EYFP; K8CreERT2 mice showed that co-deletion of Sox9 partially rescues the increased levels of K14 observed in LATS1/2-deleted luminal mammary epithelial cells. However, our analyses suggest that Sox9 and YAP/TAZ are regulating distinct transcriptional programs in this context. This conclusion is supported by examination of directly regulated target genes for YAP/TAZ and Sox9 in our LATS1/2-deletion transcriptional signatures, which showed very little

overlap in targets (see Supplementary Fig 4H and detailed further below in our responses to the comments). Our efforts with co-immunoprecipitation experiments failed to detect an association between Sox9 and YAP/TAZ, and while it is difficult to conclude from negative data, these observations support a lack of direct co-regulation of transcriptional target genes. Our conclusion from our new analyses is that YAP/TAZ activate a basal-like overgrowth phenotype in Sox9-positive cells observed following LATS1/2-deletion in luminal mammary epithelium and that Sox9 drives an independent gene expression program that partially contributes to the phenotypes observed.

2. Since YAP1 and WWTR1 are transcriptional regulators, it would have been helpful to identify transcriptional targets that govern the luminal to basal plasticity by performing chromatin immunoprecipitation experiments including SOX9 as well.

Response:

We agree that mapping the direct transcriptional targets regulated by YAP/TAZ and Sox9 following LATS1/2-deletion would be helpful to the story. In attempts to address this, we performed CUT&RUN sequencing on sorted EYFP+ cells from Isl-EYFP; K8CreERT2 mice and from LATS1/2^{ff}; Isl-EYFP; K8CreERT2 mice. We attempted to do this for YAP, TAZ, Sox9, and TEAD4, and performed two independent experiments. Despite these efforts, we were unable to attain data with sufficient quality to make meaningful conclusions on the genes are regulated by these individual factors (see Reviewer Figure 1 below).

In lieu of CUT&RUN profiling, we utilized previously described targets, identified by merging ChIP-seq and gene expression data, for YAP/TAZ (acquired from MDA-231 cells [Zanconato et al. Nat Cell Bio, 2015. PMID: 26258633]) and Sox9 (derived from basal cell carcinoma [Larsimont et al. Cell, 2015. PMID: 26095047]) to determine the extent of which our LATS1/2-deletion signatures are driven by YAP/TAZ and Sox9, respectively. We found a strong enrichment for direct targets

of YAP/TAZ in our LATS signature and modest enrichment for direct targets of Sox9. These results are now displayed in Supplementary Figure S4F-G. An analysis of the relevant transcriptional targets for YAP/TAZ and Sox9 showed little overlap in target genes regulated by these factors (Supplementary Figure S4H), suggesting divergent transcriptional regulation by these factors.

We also performed Binding Analysis for Regulation of Transcription (BART) with the top 500 upregulated and top 500 downregulated genes in our LATS KO signatures, which is a computational tool that integrates public ChIP-seq data for predicting transcription factor association of genes (Wang et al., Bioinformatics, 2018. PMID: 29608647). This analysis revealed that TEADs and YAP are among the top enriched factors associated with the top 500 upregulated genes in our LATS signature (Supplementary Table S3). These data further support a central role for YAP/TAZ and Sox9 in the transcriptional changes observed following LATS1/2 knockout.

3. It was previously suggested that luminal progenitor cells with BRCA1 mutation, could be the cell of origin for basal like breast cancers (also referenced by the authors). How does YAP1/WWTR1 correlate to BRCA1? or LATS1/2 deletion in mature luminal correlate to Basal like breast cancers that arise as a result of BRCA1 mutation. Any suggested mechanisms? based on their gene expression analysis and GSEA analysis.

Response:

To examine potential correlation with phenotypes associated with BRCA1 mutations, we compared our LATS1/2-deletion gene expression signature with that from a BRCA1-driven model of mouse breast cancer. For this, we derived signatures from the Blg-Cre; Brca1 f/f; P53f/+ mouse model of basal-like breast cancer using data from a previous study (Molyneux et al., Cell Stem Cell 2010. PMID: 20804975). We found a positive enrichment for our LATS1/2-deletion signature in the Brca1f/f-upregulated signature, and a negative enrichment in the Brca1f/f-downregulated signature. These results are now displayed in Supplementary Figure 4C. This new analysis indicates that mouse basal-like mammary carcinomas driven by Brca1-deletion show similar properties to our carcinomas driven by LATS1/2 deletion, pointing to a potential role for Hippo inactivation and/or YAP/TAZ activation in promoting tumorigenesis downstream of Brca1 loss.

4. Hippo pathway was previously shown to contribute to metastasis. Mechanistically, how does SOX9 and YAP1/WWTR1 function together in this context? The authors have made an attempt and utilized GSEA analysis to compare LATS up-regulated signature to SOX9 up-regulated signature in S4E. The selected genes as shown in heatmap from S4C and S4F has distinct genes enriched. Detailed analysis of the intersection would increase the strength of the manuscript.

Response:

As shown in our original figure S4E (new Supplementary Figure 4I), we demonstrate that in human tumors there is a positive correlation between our LATS upregulated gene expression signature and genes elevated in cells that have high Sox9 levels (Sox9^{hi}) isolated from mouse mammary basal-like carcinomas (Christin et al., Cell Reports, 2020. PMID: 32521267). Our conclusion from this data that there is a relationship between dysregulated LATS1/2 activity and Sox9 levels in human breast cancers. However, this data alone does not inform about any contributions to metastasis.

To test the question raised by the reviewer we better examined genes regulated by Sox9 in LATS1/2-Sox9 co-deleted luminal mammary epithelial cells (differentially expressed genes comparing EYFP-positive cells isolated from LATS1/2f/f;Sox9f/f;Isl-EYFP;K8CreERT2 and

LATS1/2f/f;Isl-EYFP;K8CreERT2 mice). This analysis included a gene set enrichment analysis of Sox9-regulated genes for hallmark gene sets (Supplementary Figure 4K) and a survival analysis in human breast cancers. The survival analysis revealed no significant differences in tumors with high versus low enrichment of Sox9-regulated signatures, suggesting that the genes regulated by Sox9 in our model do not correlate with more aggressive disease (Reviewer Figure 2).

Reviewer Figure 2. Kaplan-Meier plot of breast cancer patients stratified based on high and low activity of genes regulated by Sox9 in LATS1/2-Sox9 co-deleted luminal mammary epithelial cells. Stratification was performed based on the median gene expression scores and these two groups were then compared for overall survival outcomes. Significance of survival difference between groups was determined using a log-rank test.

To better examine the relationship between YAP/TAZ and Sox9, we also performed an analysis of YAP/TAZ and Sox9 direct target genes (as described in response to point 2 above), which revealed distinct transcriptional targets in LATS1/2-deleted signatures. This analysis suggests that both YAP/TAZ and Sox9 contribute to the transcriptional program downstream of LATS1/2 in mammary epithelial cells, which is supported by our observations in our genetic mouse models.

Reviewer #2

The paper “Inactivation of LATS1/2 drives luminal-basal plasticity to initiate basal-like mammary carcinomas” written by Kern et al reported that LATS1/2 knockout in mature mammary luminal epithelial cells drove the development of basal-like breast cancers. Notably, gene expression analyses of LATS1/2-deleted mammary epithelial cells revealed a transcriptional program that associates with human basal-like breast cancers. In general, the experiments and data analysis were conducted to a high standard and the findings are compelling. Figures are clearly laid out and the manuscript is well written. Since this is an area of controversy this work adds more clarity to the role of Lats (and Hippo signaling) in basal breast cancer:

We thank the reviewer for their positive assessment and have addressed all the raised points outlined in the comments below.

Comments:

1. In Fig. 1D and 2C, the IF data displayed K8/K14 and K14/YFP, respectively. The authors should consider showing K8 and YFP colocalization to indicate that the cells expanding in mammary ducts are YFP+ cells. This would be a more impactful way to show the data.

Response:

Thank you for this suggestion. We have now added a panel to show K8/YFP in the figure.

2. In Figure 1e and sFig1b, please clarify if the reduction in Sca1-expressing cells resulted from low EYFP+ cells?

Response:

We apologize that this was not made clear. In Fig 1E we are measuring Sca1 expression only in EYFP+ cells. The EYFP gate shown in Supplementary Fig 1B is drawn to encompass all EYFP-expressing (Cre+) cells. The position of cells within the EYFP gate is not expected to reflect differences in cellular phenotypes due to the binary nature of this Cre reporter.

3. Is there any relationship between YAP or YAP targets and basal-like cell markers (basal cell state)?

Response:

To further explore YAP/TAZ target genes in the context of our study, we examined a gene expression signature of YAP/TAZ-induced genes identified in MDA-MB-231 cells (Zanconato et al. Nat Cell Bio, 2015. PMID: 26258633) and performed GSVA for correlation with normal mammary cell gene sets as we did for our LATS signatures in Fig 4C. Now shown in new Supplementary Figure 4E, this analysis revealed strong enrichment for YAP/TAZ targets in normal mammary basal cells, like that of LATS1/2 deletion.

4. Please comment regarding SOX9 expression in human basal-like breast cancers? Is there any overlap with low Lats1/2 or high YAP activity?

Response:

Sox9 has been shown in prior work to be more highly expressed in human basal-like breast cancers relative to other breast cancer subtypes (Christin et al., Cell Reports, 2020. PMID: 32521267), which we have referenced in our manuscript. To specifically address the correlation between Sox9 expression and low LATS1/2 or high YAP/TAZ activity in human cancers, we performed ssGSEA in human breast cancers from TCGA. This analysis is shown in new Supplementary Figure 4J and revealed a correlation between our LATS KO-upregulated signature and Sox9 gene expression levels across all TCGA breast cancer subtypes, with both most highly enriched in the basal-like subtype. This reflects that high YAP/TAZ activity and high Sox9 expression correlate with human basal-like breast cancers relative to other subtypes.

5. Are K8 and SOX9 co-expressed in same luminal cells? Is there any difference between Lats1/2f/f; Isl-EYFP; K8CreERT2 and Lats1/2f/f; Isl-EYFP; Sox9CreERT2 mice (such as cell proliferation speed and K8/K14+ cells)?

Response:

K8 is expressed in all mammary luminal epithelial cells and Sox9 is expressed in a subset of luminal cells. The pattern of expression can be observed in Figs 2A, 2E, and S2C. The different subpopulations within the luminal mammary epithelium have been further studied elsewhere (Wang et al., Cell Reports, 2017. PMID: 27653681).

The Lats1/2f/f; Isl-EYFP; Sox9CreERT2 mice showed a rapid health decline compared with the Lats1/2f/f; Isl-EYFP; K8CreERT2 mice, necessitating earlier end points with those mice in our study.

We have added an explanation of this in the methods. To better compare the two mouse models, we performed a time course using the *Lats1/2^{f/f}; Isl-EYFP; K8CreERT2* mice, examining phenotypes daily post tamoxifen treatment (new Supplementary Figure 2B). Our observations suggest similar phenotypes between the models and support our conclusion that it is the Sox9 expressing population that is expanding in the *K8CreERT2* model.

Minor:

1. "We next sought to relay our findings to human breast cancers." Should be relate

Response:

Thank you, we have made this correction.

Reviewer #3

This paper is interesting and relevant to mammary cancer biology and its intersection with Hippo signaling. For the most part it is well written and conclusions are appropriate to the data. Several statements would benefit from more information, clarification and/or quantification. In addition, while the mammary tumor experiments with the various TG mouse models are undeniably elegant, there should be more restraint shown when conclusions are made directly relevant to breast cancers. For instance, loss of LATS1/2 in normal finite human mammary epithelial cells (as in not MCF10A or 184A1) is not sufficient to cause transformation, whereas in these mouse experiments loss of LATS1/2 results in mammary tumors. There are clearly barriers to progression that are missing in mice, and the interpretation should reflect those evolutionary differences.

Response:

We thank the reviewer for their constructive comments. We agree that differences likely exist between mouse and human mammary epithelial cells with respect to Hippo pathway regulation and function. However, we would like to highlight that, to our knowledge, no human model has recapitulated the experimental design of our study and successfully inactivated Hippo in homeostatic luminal mammary epithelial cells. We are unsure of the reviewer's specific references regarding loss of LATS1/2 in MCF10A and 184A1 cells. However, it is known that these cells express basal features (Kao et al., PLOS One 2009. PMID: 19582160). Other studies have deleted LATS1/2 in luminal breast cancer cell lines, which are likewise not directly relatable to homeostatic luminal mammary epithelial cells. Additionally, none of these systems recapitulates the complexity of the mammary microenvironment *in vivo*, which we believe informs our phenotypes as well. Due to these variables, we would not expect our models to fully recapitulate any previously published work with human cells. It is unclear at this time whether LATS1/2 deletion in homeostatic human luminal cells would promote the same phenotype we observed but performing such an experiment is technically challenging given that tools for induced luminal inhibition of both LATS1 and LATS2 are not available. Therefore, while we agree that caveats to our models do exist, we believe that our study represents an important step in understanding Hippo signaling in the homeostatic mammary epithelium *in vivo*. We have added these points to the discussion.

Figure 1, Supp. Figure 1 and accompanied text

-The reader may benefit if the text noted that phosphorylated LATS1/2 are generally thought to inactivate YAP/TAZ function.

Response:

Thank you for the suggestion. We now reference this in the text.

-General lack of quantification for IF images. How many mice and sections were looked at? Recommend quantification of specific IF images to make key points.

Response:

We agree that quantifying our images will improve the rigor of our study and have now included figures quantifying the images shown in Figs 1D (quantification in 1E), 2C (quantification in 2D), 2I (quantification in 2J), and 3C (quantification in 3D).

-Appreciate the rigor of using 2 ex vivo organoid culture models to test LATS1/2 function in other contexts. Also appreciate addressing tamoxifen treatments as a confounding factor.

Response:

Thank you for your appreciation of our efforts.

-Supp. Figure 1H

Appreciate the controls testing if Cre is only active in K8 luminal cells. It is a bit difficult to tell. This would be enhanced by including an image that is zoomed in, or perhaps through quantification.

Response:

We agree and have now added a better-quality image for this panel with a zoomed in subset to demonstrate the lineage trace in K8+ cells.

-Figure 1J- Provide a more zoomed in picture. Or some sort of quantification. It is a bit difficult to find K8/K14/tdtomato cells.

Response:

We agree and have added a more zoomed-in panel to better visualize individual K8+K14+ cells.

-Please include important information: How many mice were used in each experiment? How many control/ko mice developed tumors? How many control/ko mice developed lesions?

Response:

Thank you for this suggestion. We have added points in the text and in the methods highlighting the number of mice we analyzed histologically to make conclusions about the phenotypes we present. It is noteworthy to emphasize that the phenotype we observe in LATS1/2^{f/f};K8CreERT2 mice in Figure 1 is very dramatic, as nearly every duct in every section examined displays DCIS.

Figure 2, Supp. Figure 2 and accompanied text

-Please include important information: How many mice were used in each experiment? How many control/ko mice developed tumors? How many control/ko mice developed lesions?

Response:

We have now added points in the text and in the methods highlighting the number of mice we analyzed histologically to make conclusions about the phenotypes we present. Like the phenotype in LATS1/2f/f; Isl-EYFP;K8CreERT2 mice, nearly every duct in every section of the LATS1/2f/f;Sox9CreERT2 mice we examined displays DCIS.

-Why do the authors think the cancers occurred more rapidly when knockout was introduced in Sox9+ cells? – generally curious about their interpretation

Response:

We are sorry that information regarding the timeline of these mice was not clear. We collected these mice earlier because they were in poor health likely due to Sox9Cre-driven LATS1/2 deletion in other tissues. We have now added these details in the methods section. To better compare the two mouse models, we performed a time course using the Lats1/2f/f; Isl-EYFP; K8CreERT2 mice, examining phenotypes daily post tamoxifen treatment (new Supplementary Figure 2B). Our observations suggest similar phenotypes between the models and support our conclusion that it is the Sox9 expression population that is expanding in the K8CreERT2 model.

-Fig 2C, why is there a lack of K14 staining in zoomed in control images?

Response:

Thank you for pointing this out. The brightness of the images was normalized to the knockout, which reduced visibility in the control image due to much lower expression of K14. We have now increased brightness to allow visualization of K14 in the control.

-General lack of quantification for IF images. How many mice and sections were looked at? Recommend quantification of specific IF images to make key points.

Response:

We have performed quantification and statistical analyses of the major phenotypes throughout our manuscript and report the number of mice that were examined for these quantifications in each genetic model.

We are grateful to the reviewer for making this comment as quantification has revealed a phenotype that we missed in the LATS1/2f/f;Sox9f/f;Isl-EYFP;K8CreERT2 mice (LATS1/2-Sox9-co-deletion model). Quantification of the levels of K14 in Figure 2I showed a reduction in K14 intensity in EYFP-expressing cells in LATS1/2f/f;Sox9f/f;Isl-EYFP;K8CreERT2 mice relative to LATS1/2f/f;Isl-EYFP;K8CreERT2 mice, suggesting that Sox9-deletion reduces levels of K14 protein in these cells. We have noted this result in the revised version of the paper.

Figure 3, Supp. Figure 3 and accompanied text

-Figure 3A- YAP/TAZ expression in control myoepithelial cells is not convincing.

Response:

We thank the reviewer for pointing this out and have replaced images showing YAP/TAZ nuclear levels in Figure 3 with better quality images in the revised manuscript.

- Figure 3 would benefit from quantification of IF images as it would help to accurately assess statements in text.

Response:

We have now performed quantifications to strengthen our conclusions in Figure 3, with quantification showing reversal of K14 expression in our LATS1/2-YAP/TAZ-KO mice relative to LATS1/2-KO mice (Fig 3D), and rescue of increased organoid size in the same model (Supplementary Fig 3E).

-How many control/ko mice had DCIS? How many mice and slices were looked at?

Response:

We have added details about the numbers of mice and tissues examined in our study. We generally sectioned through the mammary gland and examined multiple sections in different regions for each experiment across multiple animals. We note that the phenotypes observed in our LATS1/2 deletion models were very striking with 100% animal penetrance and almost every duct showing a phenotype. We have also quantified the phenotypes in the LATS1/2-YAP/TAZ co-deletion model, which show a near-complete rescue (Fig 3D).

-Supp. Figure 3b-3c, It would be nice to have a zoomed in image of just YFP and just YAP in the same image to assess what is said in the text “reliable deletion of YAP and TAZ”.

Response:

We agree and have added new panels to illustrate YFP and YAP and TAZ more clearly.

-Quantify size of organoids. It would be nice to quantify the size of the organoids to assess for statistically significant morphological changes between conditions.

Response:

We agree that quantifying the size of organoids in our experiments will better define the morphological changes between the conditions we have tested. We have now added new panels displaying size quantifications for all organoid experiments we display. Please see Figures S1F, S1H, S2F, and S3E.

Regarding Figure 4, Supp Figure 4 and accompanied text

-Reasonable and supportive of findings found in previous figures.

Response:

Thank you for the positive assessment.

REVIEWERS' COMMENTS

Reviewer #1 (Remarks to the Author):

The authors have responded to all of my comments.

Reviewer #2 (Remarks to the Author):

concerns adequately addressed - shed light on multiple aspects of Lats in cancer and also connect to Sox9

Reviewer #3 (Remarks to the Author):

The authors addressed my critiques.

Response to the Reviewers' comments (response in blue)

Reviewer #1 (Remarks to the Author):

The authors have responded to all of my comments.

Reviewer #2 (Remarks to the Author):

concerns adequately addressed - shed light on multiple aspects of Lats in cancer and also connect to Sox9

Reviewer #3 (Remarks to the Author):

The authors addressed my critiques.

We thank the reviewers for their positive assessment of our revised manuscript